# Rot-Pro: Modeling Transitivity by Projection in Knowledge Graph Embedding

**Tengwei Song, Jie Luo,**∗ **Lei Huang**
State Key Laboratory of Software Development Environment
Beihang University, Beijing, 100191
{songtengwei,luojie,huangleiai}@buaa.edu.cn

## Abstract

Knowledge graph embedding models learn the representations of entities and relations in the knowledge graphs for predicting missing links (relations) between entities. Their effectiveness are deeply affected by the ability of modeling and inferring different relation patterns such as symmetry, asymmetry, inversion, composition and transitivity. Although existing models are already able to model many of these relations patterns, transitivity, a very common relation pattern, is still not been fully supported. In this paper, we first theoretically show that the transitive relations can be modeled with projections. We then propose the Rot-Pro model which combines the projection and relational rotation together. We prove that Rot-Pro can infer all the above relation patterns. Experimental results show that the proposed Rot-Pro model effectively learns the transitivity pattern and achieves the state-of-the-art results on the link prediction task in the datasets containing transitive relations.

## 1 Introduction

Knowledge graph embedding (KGE) aims to learn low-dimensional dense vectors to express the entities and relations in the knowledge graphs (KG). It is widely used in recommendation system, question answering, dialogue systems [5, 17, 7]. The general intuition of KGE is to model and infer relations between entities in knowledge graphs, which has complex patterns such as symmetry, asymmetry, inversion, composition, and transitivity as shown in Table 1.

Many studies dedicate to find a method, which is able to model various relation patterns [3, 2, 21, 20]. TransE models relations as translations, aims to model the inversion and composition patterns; DisMult can model symmetric relations by capturing interactions between head and tail entities and relations. One representative model proposed recently is RotatE [20], which is proved to be able to model symmetry, asymmetry, inversion and composition patterns by modeling relation as a rotation in the complex plane. However, none of them can model all the five relation patterns, especially the transitivity pattern.

This paper focus on modeling the transitivity pattern. We theoretically show that the transitive relations can be modeled with idempotent transformations, i.e. projections [25]. Any projection matrix is similar to a diagonal matrix with elements in the diagonal being 0 or 1. We design the projection by constraining the similarity matrix to be a rotation matrix, which has less parameters to learn.

In order to model not only transitivity but also other relation patterns shown in Table 1, we propose the Rot-Pro model which combines the projection and relational rotation together. We theoretically prove that Rot-Pro can infer the symmetry, asymmetry, inversion, composition, and transitivity

---

∗Corresponding author.

35th Conference on Neural Information Processing Systems (NeurIPS 2021).

Table 1: Common relation patterns.

| Relation pattern | Definition |
|---|---|
| Symmetry | if $(h, r, t)$, then $(t, r, h)$ |
| Asymmetry | if $(h, r, t)$, then $\neg(t, r, h)$ |
| Inversion | if $r = p^{-1}$ and $(h, r, t)$, then $(t, p, h)$ |
| Composition | if $(r = r_1 \circ \cdots \circ r_n) \wedge (h, r_1, u_1) \wedge (u_1, r_2, u_2)$ $\ldots \wedge (u_{n-1}, r_n, t)$, then $(h, r, t)$ |
| Transitivity | if $(a, r, b)$ and $(b, r, c)$, then $(a, r, c)$ |

Table 2: The supported relation patterns of several models [20].

| | Symmetry | Asymmetry | Inversion | Composition | Transitivity |
|---|---|---|---|---|---|
| TransE | ✗ | ✓ | ✓ | ✓ | ✗ |
| DistMult | ✓ | ✗ | ✗ | ✗ | ✗ |
| ComplEx | ✓ | ✓ | ✓ | ✗ | ✗ |
| RotatE | ✓ | ✓ | ✓ | ✓ | ✗ |
| Rot-Pro | ✓ | ✓ | ✓ | ✓ | ✓ |

patterns. Experimental results show that the Rot-Pro model can effectively learn the transitivity pattern. The Rot-Pro model achieves the state-of-the-art results on the link prediction task in the Countries dataset containing transitive relations and outperforms other models in the YAGO3-10 and FB15k-237 dataset.

## 2   Related work

There are mainly two types of knowledge graph embedding models, either using translation transformation or linear transformation between head and tail entities.

**Trans-series models.**   Trans-series models, which is well-known in KGE area, are essentially translation transformation based models. TransE [3] proposed a pure translation distance-based score function, which assumes the added embedding of head entity $h$ and relation $r$ should be close to the embedding of tail entity $t$. This simple approach is effective in capturing composition, asymmetric and inversion relations, but is hard to handle the 1-to-N, N-to-1 and N-N relations.

To overcome these issues, many variants and extensions of TransE have been proposed. TransH [26] projects entities and relations into a relation-specific hyperplanes and enables different projections of an entity in different relations. TransR [28] introduces relation-specific spaces, which builds entity and relation embeddings in different spaces separately. TransD [9] simplifies TransR by constructs dynamic mapping matrices. For the purpose of model optimization, some models relax the requirement for translational distance. TransA [10] replaces Euclidean distance by Mahalanobis distance to enable more adaptive metric learning, and a recent model TransMS [18] transmits multi-directional semantics by complex relations.

The variants of TransE improve the capability of the models to handle 1-to-N, N-to-1 and N-N relations as well as effectively modeling symmetric and asymmetric relations, but they are no longer able to model composition and inversion relations as they do linear transformation on head and tail entities separately before modeling the relation as translation. BoxE [1], a recent trans-series model, embeds entities as points, and relations as a set of boxes, for yielding a model that could express multiple relation patterns including composition and inversion, but it cannot express transitivity.

**Bilinear models.**   Models of bilinear series model relations as linear transformation matrix $M_r$ from head to tail entity. The type of relation patterns that a linear transformation based model can infer depends on the property of $M_r$. RESCAL [14] proposes the transformation of relation as a matrix that models the pairwise interactions between entities and relation. The score of a fact is defined by a bilinear function: $f_r = h^T M_r t$. DistMult [2] simplifies RESCAL by restricting $M_r$ to diagonal matrices. Therefore, it cannot handle other types of relations except symmetry. HolE

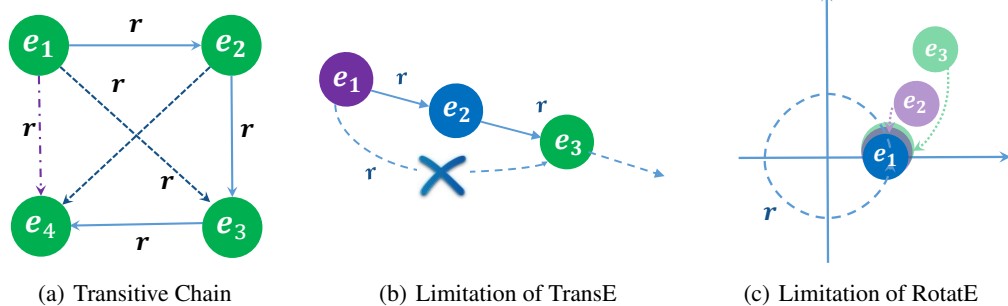

(a) Transitive Chain        (b) Limitation of TransE        (c) Limitation of RotatE

Figure 1: Illustration of transitive chain and the limitation of TransE and RotatE on representing transitivity pattern.

[12] combines the expressive power of RESCAL with the simplicity of DistMult which introduces circular correlations. HolE can express multiple types of relations, since cyclic correlation operations are not commutative.

Recently, some KGE models begin to model relation patterns explicitly. Dihedral [27] models relations in KGs with the representation of dihedral group that has properties to support the relation as symmetry. To expand Euclidean space, ComplEx [21] firstly introduces complex vector space which can capture both symmetric and asymmetric relations. RotatE [20] also models in complex space and can capture additional inversion and composition patterns by introducing rotational Hadamard product. QuatE [29] extends RotatE, using a quaternion inner product and gains more expressive semantic learning capability. ATTH [4] proposes a low-dimensional hyperbolic knowledge graph embedding method, which capture logical patterns such as symmetry and asymmetrical.

However, none of existing models is capable of modeling transitivity relation pattern. We are the first to show that the transitivity can be modeled with projections, and we prove that the proposed model is able to infer all the five relation patterns shown in Table 2.

## 3 Rot-Pro: Modeling Relation Patterns by Projection and Rotation

### 3.1 Preliminary

RotatE is a representative approach that models a relation as an element-wise rotation from the embedding $\mathbf{e}_h$ of head entity to the embedding $\mathbf{e}_t$ of tail entity in complex vector space. This can be denoted as $\mathbf{e}_t(k) \approx rot(\mathbf{e}_h(k), \theta_r(k))$, where $\theta_r$ is the embedding of relation $r$ and $rot(\mathbf{e}_h(k), \theta_r(k))$ is the rotation function that rotates the $k$th element of $\mathbf{e}_h$ with a phase of the $k$th element of $\theta_r$. For each embedding $\mathbf{e}$, let $Re(\mathbf{e}(k))$ and $Im(\mathbf{e}(k))$ be the real number and imaginary number of its $k$th dimension, the rotation transformation in the $k$th dimension is defined by an orthogonal matrix whose determinant is 1, as follows:

$$\begin{bmatrix} Re(\mathbf{e}_t(k)) \\ Im(\mathbf{e}_t(k)) \end{bmatrix} \approx \begin{bmatrix} Re(rot(\mathbf{e}_h(k), \theta_r(k))) \\ Im(rot(\mathbf{e}_h(k), \theta_r(k))) \end{bmatrix} = \begin{bmatrix} \cos\theta_{\mathbf{r}}(k) & -\sin\theta_{\mathbf{r}}(k) \\ \sin\theta_{\mathbf{r}}(k) & \cos\theta_{\mathbf{r}}(k) \end{bmatrix} \begin{bmatrix} Re(\mathbf{e}_h(k)) \\ Im(\mathbf{e}_h(k)) \end{bmatrix}. \quad (1)$$

It has been proved that RotatE can infer symmetry, asymmetry, inversion, and composition relation patterns [20]. However, it cannot infer the transitivity pattern, and we will explain in what follows.

### 3.2 Representation of transitive relation

Relation $r$ is a transitive relation, if for any instances $(e_1, r, e_2)$ and $(e_2, r, e_3)$ of relation $r$, $(e_1, r, e_3)$ is also an instance of $r$. For convenience of illustration, we define the transitive chain of a transitive relation as follows.

**Definition 1.** *A* transitive chain *of $r$ is defined as a chain of instances $(e_1, r, e_2), \ldots, (e_{m-1}, r, e_m)$ of $r$, where $e_1, \ldots, e_m$ are different entities.*

For the transitive closure of a transitive chain, every two entities in the chain should be connected by the transitive relation $r$. Hence, it can be represented as a fully connected directed graph with $\frac{m(m-1)}{2}$ edges. It can be proved that a transitive relation can be represented as the union of transitive closures of all transitive chains. Thus, the representation of transitive relations can be reduced to the representation of transitive chains.

An example of a transitive chain and its transitive closure are shown in Figure 1(a), where $(e_1, r, e_2), (e_2, r, e_3), (e_3, r, e_4)$ form a transitive chain, $(e_1, r, e_3)$ and $(e_2, r, e_4)$ are instances derived by transitivity via one-hop, and $(e_1, r, e_4)$ is the only instance derived via two-hops.

Due to the speciality of transitivity, current models are unable to effectively model such transformation in vector space. For instance, in TransE (Figure 1(b)), where a relation is regarded as a translation between the head and tail entities, it requires the translation to be a zero vector to model transitivity, which forces the embeddings of entities in a transitive chain to be the same. Thus, it cannot model transitivity. In RotatE (Figure 1(c)), it requires the relational rotation phase $\theta_r(k)$ in each dimension to be $2n\pi$ ($n = 0, 1, \ldots$) to model transitivity, which also forces the embeddings of entities to be the same in a transitive chain.

**Our solution.** Based on the observation on transitive closures of transitive chains, in each transitive chain $(e_1, r, e_2), \ldots, (e_{m-1}, r, e_m)$, for each entity $e_j$, $(e_j, r, e_{j+l})$ can be derived by transitivity via $l - 1$-hops ($2 \leqslant l \leqslant m - j$). If we model each relation $r$ as a kind of transformation $T_r$, then it requires $T_r(\mathbf{e}_h) = \mathbf{e}_t$ for each relation instance $(h, r, t)$. Therefore, the transformation of a transitive relation must satisfies that $T_r^l(\mathbf{e}_j) = T_r(\mathbf{e}_j)$ ($1 \leqslant j \leqslant m, 1 \leqslant l \leqslant m - j$), i.e. the result of transforming an entity embedding multiple times is equivalent to that of transforming it once. This inspires us to model the transitivity pattern in terms of the idempotent transformation (projection [25]) which has the same property. For each relation $r$, let $S_r(k)$ be an invertible matrix on the $k$th dimension, a general orthogonal projection is defined by the idempotent matrix:

$$M_r(k) = S_r(k)^{-1} \begin{bmatrix} a_r(k) & 0 \\ 0 & b_r(k) \end{bmatrix} S_r(k), \quad (a_r(k), b_r(k) \in \{0, 1\}). \tag{2}$$

Without loss of generality, we simply set $S_r(k) = \begin{bmatrix} \cos\theta_p(k) & -\sin\theta_p(k) \\ \sin\theta_p(k) & \cos\theta_p(k) \end{bmatrix}$ to be a rotation matrix, which rotates the original axis by a phase $\theta_p(k)$. The orthogonal projection $p_r(k)$ defined by $M_r(k)$ is performed in the new axis after rotation:

$$p_r(k)(x + yi) = [1\ i]M_r(k)\begin{bmatrix} x \\ y \end{bmatrix}. \tag{3}$$

In the rest of paper, we will omit the dimensional indices $(k)$ in $M_r(k)$ and $p_r(k)$ for simplicity. In this way, for entities $e_1, \ldots, e_m$ in a transitive chain, we have $p_r^l(\mathbf{e}_j(k)) = p_r(\mathbf{e}_j(k))$ ($1 \leqslant j \leqslant m, 1 \leqslant l \leqslant m - j$). This implies that $p_r(\mathbf{e}_1(k)) = \cdots = p_r(\mathbf{e}_m(k))$, which does not force the entity embeddings $\mathbf{e}_1, \ldots, \mathbf{e}_m$ to be the same. The embeddings $\mathbf{e}_1, \ldots, \mathbf{e}_m$ can be different to each other and have the same projected vector under $p_r$.

### 3.3 The Rot-Pro model

**Model formulation.** In order to model not only transitivity but also other relation patterns shown in Table 1, we combine the above projection based representation for transitivity and the relational rotation based representation for symmetry, asymmetry, inversion, and composition together. We propose Rot-Pro to model relations as relational rotations on the projected entity embeddings on complex space $\mathbb{C}^d$. For each triple $(h, r, t)$, the Rot-Pro model requires that

$$rot(p_r(\mathbf{e}_h(k)), \theta_r(k)) = p_r(\mathbf{e}_t(k)). \tag{4}$$

We demonstrate in the following theorem that Rot-Pro enables the modeling and inferring of all the five types of relation patterns introduced above.

**Theorem 1.** *Rot-Pro can infer the symmetry, asymmetry, inversion, composition, and transitivity patterns.*

*Proof.* (1) Let $a_r = 1$ and $b_r = 1$, $M$ becomes an identity matrix and $p$ becomes an identity transformation, and our model is reduced to the RotatE model. Thus, Rot-Pro can also infer the symmetry, asymmetry, inversion and composition patterns as RotatE does [20].

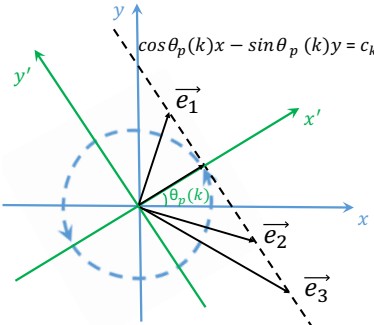

Figure 2: The representation of transitivity pattern in complex plane.

(2) Here we will construct the solutions in Rot-Pro model for transitive relations. Let $a_r = 1$ and $b_r = 0$, $p_r$ is a projection to the real axis $x'$ as shown in Figure 2 [2]. As discussed in previous section, to model the transitivity of relation $r$, the projected entity embeddings in a transitive chain must satisfy that $rot(p(rot(p(\mathbf{e}_j(k)), \theta_r(k))), \theta_r(k)) = rot(p_r(\mathbf{e}_j(k)), \theta_r(k)) = p_r(\mathbf{e}_{j+2}(k)) = p_r(\mathbf{e}_{j+1}(k))$. Therefore, the phase of relational rotation $\theta_r(k)$ can only be $2n\pi$ $(n = 0, 1, 2, \ldots)$ and $p_r(\mathbf{e}_j(k)) = p_r(\mathbf{e}_1(k))$ for any $1 < j \leqslant m$.

According to Equation 4, the following equation is expected to hold.

$$M_r \begin{bmatrix} Re(e_j(k)) \\ Im(e_j(k)) \end{bmatrix} = \begin{bmatrix} \cos\theta_r(k) & -\sin\theta_r(k) \\ \sin\theta_r(k) & \cos\theta_r(k) \end{bmatrix} \left( M_r \begin{bmatrix} Re(e_1(k)) \\ Im(e_1(k)) \end{bmatrix} \right) = M_r \begin{bmatrix} Re(e_1(k)) \\ Im(e_1(k)) \end{bmatrix} \quad (5)$$

The above equation holds iff

$$\cos\theta_p(k)Re(e_j(k)) - \sin\theta_p(k)Im(e_j(k)) = \cos\theta_p(k)Re(e_1(k)) - \sin\theta_p(k)Im(e_1(k)). \quad (6)$$

This equation holds if for any $e_j$ in the transitive chain,

$$\cos\theta_p(i)Re(e_j(k)) - \sin\theta_p(k)Im(e_j(k)) = c_k,$$

where $c_k = \cos\theta_p(k)Re(e_1(k)) - \sin\theta_p(k)Im(e_1(k))$ is a constant. That is, all these entity embeddings $\mathbf{e}_j(k) = x + yi$ in the transitive chain are located in the line defined by Equation (7) on the complex plane as shown in Figure 2.

$$\cos\theta_p(k)x - \sin\theta_p(k)y = c_k. \quad (7)$$

Here, different value of $c_k$ can represent different transitive chain. In summary, we construct the solutions for representing transitivity in Rot-Pro model, i.e. $a_r(k) = 1$, $b_r(k) = 0$, $\theta_r(k) = 2n\pi$, and for any entity embedding $\mathbf{e}_j$, it satisfies that $\cos\theta_p(i)Re(e_j(k)) - \sin\theta_p(k)Im(e_j(k)) = c_k$, where $c_k$ is a constant. $\qquad \square$

**Score function.** For each triple $(h, r, t)$, the distance function of the Rot-Pro model is defined as following:

$$d_r(\mathbf{e}_h, \mathbf{e}_t) = \|rot(p_r(\mathbf{e}_h), \theta_r) - p_r(\mathbf{e}_t)\|. \quad (8)$$

The score function $f_r(\mathbf{e}_h, \mathbf{e}_t) = -d_r(\mathbf{e}_h, \mathbf{e}_t)$.

### 3.4 Optimization objective

In the training process, we adopt the self-adversarial negative sampling, which has been proved as an effective optimization approach to KGE [20]. The negative sampling loss $\mathcal{L}_s$ with self-adversarial training is defined as:

$$\mathcal{L}_s = -\log\sigma(\gamma - d_r(\mathbf{h}, \mathbf{t})) - \sum_{j=1}^{n} p(h'_j, r, t'_j)\log\sigma(d_r(\mathbf{h}'_j, \mathbf{t}'_j) - \gamma) \quad (9)$$

---

[2]We can also set $a = 0$ and $b = 1$, then $p$ is a projection to the imaginary axis $y'$.

where $\gamma$ is a fixed margin, $\sigma$ is the sigmoid function, $(h'_j, r, t'_j)$ is the $j$th negative instance and $p(h'_j, r, t'_j)$ is the distribution for negative sampling [20].

In addition, to ensure the learned matrix to be a projection, the values of $a$ and $b$ in Equation 2 should be restricted to 0 or 1. To enforce such constraint, we proposed a projection penalty loss as follows:

$$\mathcal{L}_p = \sum_{j=1}^{|R|} (||(\mathbf{a}_j - 1.0) \odot \mathbf{a}_j \odot \mathbf{q}_j||_2 + ||(\mathbf{b}_j - 1.0) \odot \mathbf{b}_j \odot \mathbf{q}_j||_2). \tag{10}$$

Here $|R|$ is the number of relations, $\odot$ is the Hadamard product, and $\mathbf{q}_j = \{q_j(k)\}_{k=1}^d$, where $q_j(k) = 1$ if $[(\mathbf{x}_j(k) - 1.0) \odot (\mathbf{x}_j(k) - 0.0)] < \gamma_m$, otherwise $q_j(k) = \beta > 1$. Here $\gamma_m$ and $\beta$ are hyper-parameters. We define $\mathbf{q}_j$ to impose more penalty to values which are far from 0 or 1 than that of values which are close to 0 or 1.

Let $\alpha$ be a hyper-parameter, the total loss is defined as the weighted average of the above two losses.

$$\mathcal{L} = \mathcal{L}_s + \alpha \cdot \mathcal{L}_p. \tag{11}$$

# 4 Experiments

## 4.1 Datasets

We evaluate the Rot-Pro model on four well-known benchmarks. In general, FB15k-237 and WN18RR are two widely-used benchmarks and YAGO3-10 and Countries are two benchmarks with abundant relation patterns including transitivity.

- **FB15k-237:** Freebase [11] contains information including people, media, geographical and locations. FB15k is a subset of Freebase and FB15k-237 [24] is a modified version of FB15k, which excludes inverse relations to resolve a flaw with FB15k [23]. It contains 14,541 entities, 237 relations, and 272,115 training triples.

- **WN18RR:** WN18RR [23] is a subset of WN18 [3] from WordNet [15]. WordNet is a dataset that characterizes associations between English words. Compared with WN18, WN18RR retains most of the symmetric, asymmetric and compositional relations, while removing the inversion relations. It contains 40,943 entities, 11 relations, and 86,835 training triples.

- **YAGO3-10:** YAGO [19] is a dataset which integrates vocabulary definitions of WordNet with classification system of Wikipedia. YAGO3-10 [13] is a subset of YAGO, which contains 123,182 entities, 37 relations and 1,079,040 training triples. According to the ontology of YAGO3, it contains almost all common relation patterns.

- **Countries:** Countries [8] is a relatively small-scale dataset, which contains 2 relations and 272 entities (244 countries, 5 regions and 23 sub-regions). The two relations of Countries are *locatedIn* and *neighborOf*, which are transitive and symmetric relations respectively. The Countries dataset has 3 tasks, each requiring inferring a composition pattern with increasing length and difficulty.

## 4.2 Evaluation protocol

We evaluate the KGE models on three common evaluation metrics: mean rank (MR), mean reciprocal rank (MRR), and top-$k$ Hit Ratio (Hit@$k$). For each valid triples $(h, r, t)$ in the test set, we replace either $h$ or $t$ with every other entities in the dataset to create corrupted triples in the link prediction task. Following previous work [3, 23, 6, 29, 16], all the models are evaluated in a *filtered* setting, i.e, corrupt triples that appear in training, validation, or test sets are removed during ranking. The valid triple and filtered corrupted triples are ranked in ascending order based on their prediction scores. Lower MR, higher MRR or higher Hit@$k$ indicate better performance.

## 4.3 Experiment setup

With the hyper-parameters introduced, we train Rot-Pro using a grid search of hyper-parameters: fixed margin $\gamma$ in Equation 9 $\in \{0.1, 4.0, 6.0, 9.0, 16.0, 20.0\}$, weights tuning hyper-parameters for loss, $\alpha \in \{0.0001, 0.0005, 0.0008\}$, value of $\gamma_m$ in Equation 10 $\in \{1e^{-6}, 5e^{-6}, 1e^{-5}\}$, value

Table 3: Link prediction results on FB15k-237 and WN18RR.

| | FB15k-237 | | | | | WN18RR | | | | |
|---|---|---|---|---|---|---|---|---|---|---|
| | MR | MRR | Hit@1 | Hit@3 | Hit@10 | MR | MRR | Hit@1 | Hit@3 | Hit@10 |
| TransE [3] | 357 | .294 | - | - | .465 | 3384 | .226 | - | - | .501 |
| DistMult [2] | 254 | .241 | .155 | .263 | .419 | 5110 | .43 | .39 | .44 | .49 |
| ComplEx [21] | 339 | .247 | .158 | .275 | .428 | 5261 | .44 | .41 | .46 | .51 |
| ConvE [23] | 244 | .325 | .237 | .356 | .501 | 4187 | .43 | .40 | .44 | .52 |
| RotatE [20] | 177 | .338 | .241 | .375 | .533 | 3340 | **.476** | **.428** | **.492** | .571 |
| BoxE [1] | **163** | .337 | .238 | .347 | .538 | 3207 | .451 | .400 | .472 | .541 |
| Rot-Pro | 201 | **.344** | **.246** | **.383** | **.540** | **2815** | .457 | .397 | .482 | **.577** |

Table 4: Link prediction results on YAGO3-10 and Countries.

| | YAGO3-10 | | | | | Countries (AUC-PR) | | |
|---|---|---|---|---|---|---|---|---|
| | MR | MRR | Hit@1 | Hit@3 | Hit@10 | S1 | S2 | S3 |
| DistMult [2] | 5926 | .34 | .24 | .38 | .54 | 1.00 | 0.72 | 0.52 |
| ComplEx [21] | 6351 | .36 | .26 | .40 | .55 | 0.97 | 0.57 | 0.43 |
| ConvE [23] | 1671 | .44 | .35 | .49 | .62 | 1.00 | 0.99 | 0.86 |
| RotatE [20] | 1767 | .495 | .402 | .550 | .670 | 1.00 | 1.00 | 0.95 |
| BoxE [1] | **1022** | **.560** | **.484** | **.608** | .691 | - | - | - |
| Rot-Pro | 1797 | .542 | .443 | .596 | **.699** | **1.00** | **1.00** | **0.998** |

of $\beta$ in Equation 10 $\in \{1.3, 1.5, 2.0\}$. Both the real and imaginary parts of the entity embeddings are uniformly initialized, and the phases of the relational rotations are initialized between $\{(-\pi, \pi), (-\frac{\pi}{2}, \frac{\pi}{2})\}$. In some settings, the phases of the relational rotations are also normalized to between $\{(-\pi, \pi), (-\frac{\pi}{2}, \frac{\pi}{2})\}$ during training.

## 4.4 Main results

We compare Rot-Pro with several state-of-the-art models, including TransE [3], DistMult [2], ComplEx [21], ConvE [23], as well as RotatE [20] and BoxE [1], to empirically show the importance of being able to model and infer more relation patterns for the task of predicting missing links. Table 3 summarizes our results on FB15k-237 and WN18RR, where results of baseline models are taken from Sun et al [20] and Ralph et al [1]. We can see that Rot-Pro outperforms the baseline models on most evaluation metrics. Compared to RotatE, the improvement of Rot-Pro is limited since there is no sufficient transitive relation defined on these two datasets, but the results are still comparable with other baseline models.

Table 4 summarizes our results on YAGO3-10 and Countries, which contain transitive relations. Hence the improvement of Rot-Pro over RotatE and other linear transformation models is much more significant. Specifically, Rot-Pro obtains better AUC-PR result than existing state-of-the-art approaches, which indicates that Rot-Pro could effectively infer relation patterns such as transitivity, symmetry and composition. As a translation transformation model, BoxE outperforms Rot-Pro on YAGO3-10 on most evaluation metrics, which indicates it is also a strong KGE model for inferring multiple relation patterns. However, the performance of BoxE on specific transitivity test sets is still not comparable with Rot-Pro, where additional experiments can be found in the appendix.

## 4.5 Validation of learned representation of transitive relations

We conduct further analysis on the the Rot-Pro model to verify that the model can actually learn the representations of transitive relations and have the theoretical property as expected. To do this, we first investigate the distributions of relational rotation phases in all dimensions of entity embeddings obtained by training on YAGO3-10. According to our theoretical analysis, we expect the model could learn to represent transitivity, i.e. for any non-trivial projection (i.e. $a = 1, b = 0$ or $a = 0, b = 1$), the corresponding phase of relational rotation should be $2n\pi$. The experimental results are shown in Figure 3(a). It can be observed that the Rot-Pro model does learn the relational rotation phases 0 and $2\pi$ as expected. However, it also learns the unexpected relational rotation phase $\pi$. Further

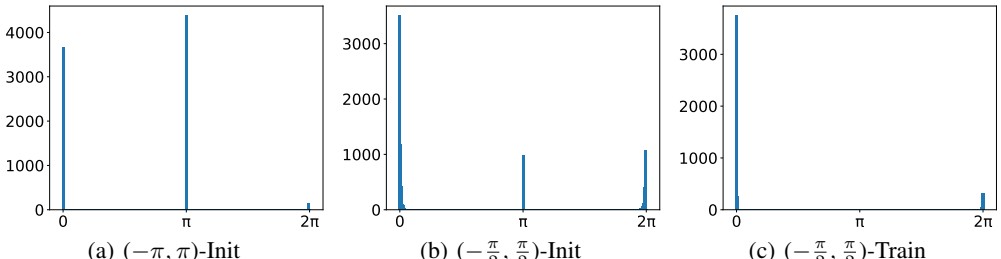

(a) $(-\pi, \pi)$-Init      (b) $(-\frac{\pi}{2}, \frac{\pi}{2})$-Init      (c) $(-\frac{\pi}{2}, \frac{\pi}{2})$-Train

Figure 3: Distributions of relational rotation phases. The $x$-axis is the relational rotation phases. The $y$-axis is the number of dimensions of relation embeddings that have non-trivial projection before rotation with a specific phase, i.e. the embedding of parameter $a$ and $b$ are $1, 0$ or $0, 1$.

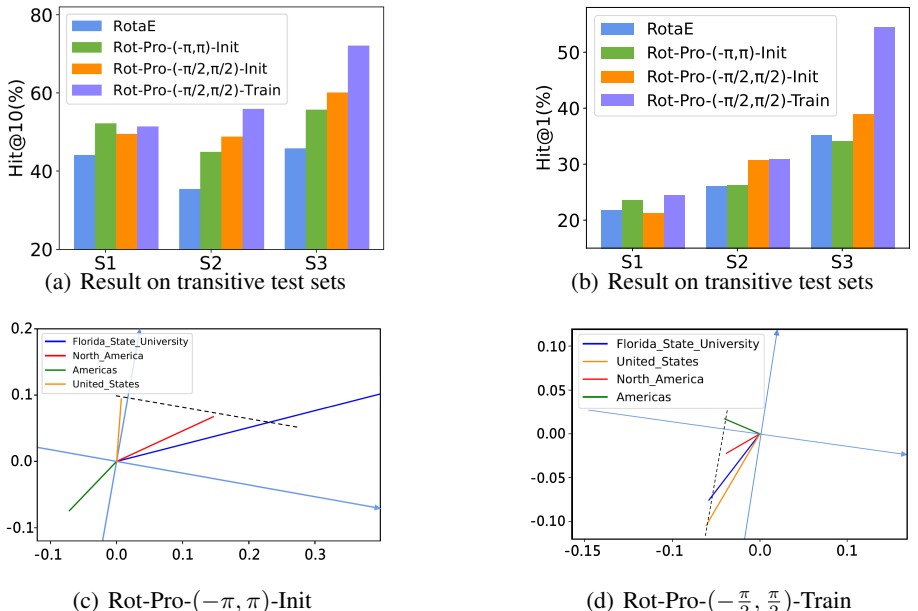

(c) Rot-Pro-$(-\pi, \pi)$-Init          (d) Rot-Pro-$\left(-\frac{\pi}{2}, \frac{\pi}{2}\right)$-Train

Figure 4: (a) shows the Hit@10 results of the RotatE and Rot-Pro models on three test sets for transitivity. (b) and (c) show the representation of four entities in a transitive chain in two variants of Rot-Pro models with different constraints of relational rotation phase.

experiments reveal that by turning the initialization range of the relational rotation phases, the problem of learning unexpected relational rotation phase $\pi$ could be mitigated. By changing the initialization range of relational rotation phases from $(-\pi, \pi)$ to $\left(-\frac{\pi}{2}, \frac{\pi}{2}\right)$, the number of relational rotation phases $\pi$ becomes significantly less. When we further restrict the relational rotation phases to $\left(-\frac{\pi}{2}, \frac{\pi}{2}\right)$ during training, almost all relational rotation phases become $0$ or $2\pi$.

The results above are also reflected in the quantitative test. To fully understand the impact of changing initialization range on the performance of the Rot-Pro model on modelling transitive relations, we construct three sub-test sets S1, S2, S3 of YAGO3-10 for evaluation, which consist of a single transitive relation *isLocatedIn*. Test set $S1$ contains instances of *isLocatedIn* in the original test set. Test set $S2$ is obtained by applying the transitivity once on instances of *isLocatedIn* in the YAGO3-10 dataset. Test set $S3$ is constructed similarly to $S2$, except by applying the transitivity at least twice. We take the RotatE model as baseline, and compare it with three variant of Rot-Pro models with different settings: the first one with relational rotation phase initialized in $(-\pi, \pi)$, the second one with relational rotation phase initialized in $\left(-\frac{\pi}{2}, \frac{\pi}{2}\right)$, the third one with relational rotation phase restricted in $\left(-\frac{\pi}{2}, \frac{\pi}{2}\right)$ during training. The experimental result is shown in Figure 4(a). It shows that tuning of initialization range also largely improves the performance of the Rot-Pro model, which

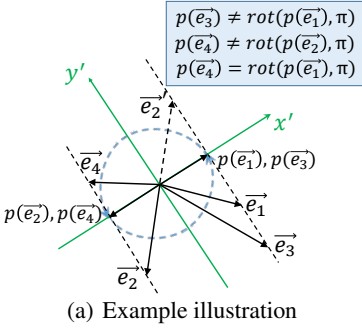
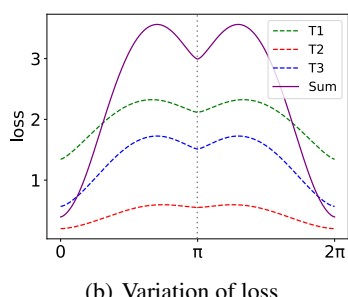

(a) Example illustration       (b) Variation of loss

Figure 5: (a) is an example of miss-placed four entities on a transitive chain, which consist of three triples: $T_1$: (Florida_State_University, isLocatedIn, United_States), $T_2$: (United_States, isLocatedIn, North_America), $T_3$: (North_America, isLocatedIn, Americas). (b) is the variation of loss for these triples.

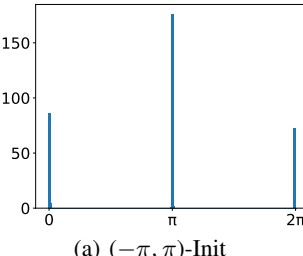
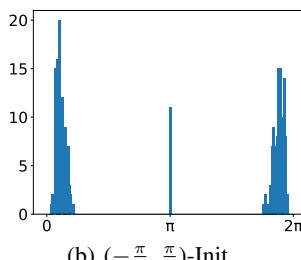
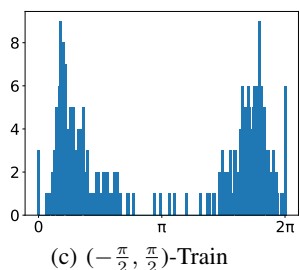

(a) $(-\pi, \pi)$-Init       (b) $(-\frac{\pi}{2}, \frac{\pi}{2})$-Init       (c) $(-\frac{\pi}{2}, \frac{\pi}{2})$-Train

Figure 6: The distribution of relational rotation phases of three Rot-Pro variants over all dimensions of a specific symmetric relation *isMarriedTo*. The meaning of $x$ and $y$ axes is the same as Figure 3.

coincides with the improvement of learning correct representations of transitive relations. All the variant of Rot-Pro models outperform RotatE significantly, especially when the relational rotation phase is restricted to $(-\frac{\pi}{2}, \frac{\pi}{2})$ during training.

We also visualize one dimension of embeddings of three entities connected by a transitive relation *isLocatedIn* in YAGO3-10. Figure 4(c) shows the visualization of entity embeddings of the Rot-Pro model trained with initialization range $(-\pi, \pi)$, which contains a miss placed entity embedding. While Figure 4(d) is the visualization of embeddings of entities of the Rot-Pro model trained by restricting the relational rotation phase to $(-\frac{\pi}{2}, \frac{\pi}{2})$ during training, where all entity embeddings in the transitive chain are placed correctly as expected. We can see that these vectors are basically fit in a line and can almost be projected to the same vector in the rotated axis.

**Explanation.** The Rot-Pro model with no additional restrictions on the relational rotation phase may learn a phase $\pi$ which does not exactly meet our expectation. A possible representations of four entities in a transitive chain with relational rotation phase $\pi$ is illustrated in Figure 5(a), in which four out of the six instances of transitive relation are correctly represented, while the other two instances $(e_1, r, e_3)$ and $(e_2, r, e_4)$ are not. Obviously, this is not an optimal solution for the model, and the reason is likely to be that the model falls into a local optimum during the learning process. To demonstrate this, we plot the variation of loss for three triples in a transitive chain with the relational rotation phase range over $(0, 2\pi)$. The result is shown in Figure 5(b). We can find that there is indeed a local optimum at $\pi$, and the global optimum is at 0 and $2\pi$, which is consistent with our conjecture.

### 4.6 Limitation

According to the experimental results, Rot-Pro is sensitive to the range of relational rotation phases, and hence prone to fall into the local optimum solution. Though it can learn the idempotency

of transitivity correctly by enforcing addition constraints on training, however, such constraints have negative impact on the learning of other relation patterns, such as symmetry. We can find in Figure 6(a) that for a symmetric relation, the relational rotation phases learned by a Rot-Pro without phase constraint are either $0$, $\pi$ or $2\pi$, which indicates that it has similar capability of modeling and inferring symmetry relation pattern as RotatE. By narrowing of the range of relational rotation phases, the histogram on the symmetry relation is gradually disrupted as shown in Figure 6(b) and 6(c). Therefore, a trade-off should be made between the better modeling and inferring of transitivity and the other relation patterns. Such limitation might be further optimized through learning each relation pattern separately and integrate through mechanisms such as attention, which we will study in future works.

## 5  Conclusion and Future Work

In this paper, we theoretically showed that the transitive relations can be modeled with projections that is an idempotent transformation. We also theoretically proved that the proposed Rot-Pro model is able to infer the symmetry, asymmetry, inversion, composition, and transitivity patterns. Our experimental results empirically showed that the Rot-Pro model can effectively learn the transitivity pattern. Our model also has the potential to be improved by extending the complex space to higher dimension space, such as quaternion space [29, 22]. While the proof of expressiveness in many previous works is focused on the expressiveness of each relation pattern separately, it is worthwhile to further investigate whether a model can handle all common relation patterns simultaneously, considering that a single relation may exhibit multiple relation patterns and different relation patterns may have complex interactions in knowledge graphs.

## Acknowledgments and Disclosure of Funding

This work was supported by the National Key Research and Development Plan of China (Grant No. 2018AAA0102301) and the National Natural Science Foundation of China (Grant No. 61690202).

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
