# Supplementary Material of Rot-Pro: Modeling Transitivity by Projection in Knowledge Graph Embedding

**Tengwei Song,    Jie Luo,**[*]   **Lei Huang**
State Key Laboratory of Software Development Environment
Beihang University, Beijing, 100191
{songtengwei,luojie,huangleiai}@buaa.edu.cn

## A   Proof of a property of transitive relation

In section 3.2 of the submitted paper, we use the conclusion that "the transitive relation can be represented as the union of transitive closures of of all transitive chains." Here, we prove it in the following lemma.

**Lemma 1.** *A transitive relation can be represented as the union of transitive closures of all transitive chains.*

*Proof.* For any given transitive relation $r$, it can be represented as a directed graph $G_r$ which satisfies that for any vertex $e$ and $e'$, if there is a path from $e$ to $e'$, then there is an edge connecting $e$ to $e'$ directly. We can see that if there is a path $e, e_1, \ldots, e_m, e'$ from $e$ to $e'$ whose length is larger than 1 (i.e. $m \geqslant 1$), then the edge connecting $e$ and $e'$ directly can be derived through transitivity, i.e. the transitive chain $(e, r, e_1), \ldots, (e_m, r, e')$ implies that $(e, r, e')$. By removing any edge $(e, r, e')$ that $e$ and $e'$ is connected by a path longer than 1, we can obtain a new graph $G'_r$. For any instance $(e, r, e')$, if it is an edge in $G'_r$, then $(e, r, e')$ is a transitive chain itself and hence it is in the transitive closure of itself; otherwise, $(e, r, e')$ is an edge removed from $G_r$ and hence there is a path from $e$ to $e'$ in $G_r$ whose length is larger than 1, then $(e, r, e')$ can be derived through transitivity based on the path, i.e. $(e, r, e')$ is in the transitive closure of the path (transitive chain). Hence any instance of a transitive relation is in a transitive closure of a transitive chain. Thus, a transitive relation can be represented as the union of transitive closures of all transitive chains. □

## B   Statistics and split of datasets.

The datasets we used for experiments are open-sourced, which can be obtained in the source code[2] of RotatE [1]. Table 1 shows the statistic of these datasets, where the number of training triples in the S1, S2, and S3 datasets of Counties are separated by '/'.

## C   Computational resources

Our model is implemented in Python 3.6 using Pytorch 1.1.0. Experiments are performed on a workstation with Intel Xeon Gold 5118 2.30GHz CPU and NVIDIA Tesla V100 16GB GPU.

---

[*]Corresponding author.

[2]https://github.com/DeepGraphLearning/KnowledgeGraphEmbedding/tree/master/data

35th Conference on Neural Information Processing Systems (NeurIPS 2021).

Table 1: Statistics of datasets

|  | entities | relations | Triples | | |
|---|---|---|---|---|---|
|  |  |  | train | valid | test |
| FB15k-237 | 14,541 | 237 | 272,115 | 17,535 | 20,466 |
| WN18RR | 40,943 | 11 | 86,835 | 3,034 | 3,134 |
| YAGO3-10 | 123,182 | 37 | 1,079,040 | 5,000 | 5,000 |
| Countries | 271 | 2 | 985/ 1,063/ 1,111 | 24 | 24 |

## D Hyper-parameter settings

We list the best hyper-parameter setting of Rot-Pro on the above datasets in Table 2. The setting of dimension $d$ and batch size $b$ is the same as RotatE [1].

Table 2: Hyper-parameter settings

|  | dimension $d$ | batch size $b$ | fixed margin $\gamma$ | $\gamma_m$ | $\beta$ | $\alpha$ |
|---|---|---|---|---|---|---|
| FB15k-237 | 1000 | 1024 | 9.0 | 0.000001 | 1.5 | 0.001 |
| WN18RR | 500 | 512 | 4.0 | 0.000001 | 1.3 | 0.0003 |
| YAGO3-10 | 500 | 1024 | 16.0 | 0.000001 | 1.5 | 0.0005 |
| Countries | 500 | 512 | 0.1 | 0.000001 | 1.5 | 0.0005 |

## E Transitivity performance comparison with BoxE

The fully expressive of BoxE refers to that it is able to express inference patterns, which includes symmetry, anti-symmetry, inversion, composition, hierarchy, intersection, and mutual exclusion. However, it does not model and infer the transitivity pattern. Therefore, we further conducted experiments on the three sub-test sets S1, S2, S3 we sampled from YAGO3-10 as described in the paper to verify this. The experimental results are listed as below.

Table 3: Link prediction result of BoxE and Rot-Pro on S1, S2, S3 test sets.

|  | S1 | | S2 | | S3 | |
|---|---|---|---|---|---|---|
|  | BoxE | Rot-Pro | BoxE | Rot-Pro | BoxE | Rot-Pro |
| MR | .343 | .337 | .290 | .328 | .381 | .447 |
| Hit@1 | .255 | .247 | .262 | .235 | .349 | .337 |
| Hit@3 | .385 | .376 | .291 | .366 | .385 | .517 |
| Hit@10 | .504 | .512 | .342 | .522 | .439 | .626 |

## References

[1] Zhiqing Sun, Zhi-Hong Deng, Jian-Yun Nie, and Jian Tang. RotatE: Knowledge graph embedding by relational rotation in complex space. In *International Conference on Learning Representations*, 2019.