# OpenReview forum: "Rot-Pro: Modeling Transitivity by Projection in Knowledge Graph Embedding"
_NeurIPS.cc/2021/Conference — NeurIPS 2021 Poster_

### Official Review · Reviewer_99hH · 2021-07-11

**Rating:** 6
**Confidence:** 4

**Summary:**

This paper is related to the problem of Knowledge Graph (KG) Embedding. It seems that none of the existing approaches for KG embedding is capable of modeling the transitivity property of the relation which is an important property for many relations to have in real-world KGs. The aim of this paper, therefore, is to bridge this specific research gap.

This paper proposes a KG embedding model called __Rot-Pro__ which not only is capable of modeling the transitive property of a relation but also other properties including symmetry, asymmetry, inversion, and composition. The approach is based on applying a projection operator on the embedding of the head and tail entities before applying RotatE so as to be able to model transitive relations. The experiments are performed on two well-known datasets FB15k-237 and WN18RR. The experiments are also performed on two other datasets called YACO3-10 and Countries because they have lots of transitive relations present in them.

On the datasets, where transitive relations are dominant, the approach seems to do justice but the approach seems to be facing some hiccups in the presence of other types of relations such as symmetry.

**Limitations And Societal Impact:**

Yes. The authors have explicitly discussed the limitations of the proposed method which they observed during experimentation. There are a few minor clarifications I have in this regard which I have listed in my points above.

**Main Review:**

__Originality__
- The problem of KG embedding is a well-known research topic in the field. This paper identifies an important limitation of the existing approaches and tries to fix that.
- The proposed approach in this paper builds on top of the well-known approach called RotatE. The proposed approach suggests adding one extra pre-processing step to the entity embeddings before calling the routine of RotatE.  The pre-processing step suggests that all the entity embeddings should be first (orthogonally) projected into a subspace of the embedding space via projection matrices. These projection matrices are specific to each relation as well as each embedding dimension. This extra step introduces only 3 extra model parameters ($\theta_p(k)$ and $a$ or $b$) per embedding dimension.
- This extra step alone allows RotatE to model the transitive property of relations. Overall, this extra step, though seemingly small, does offer the desired effect (at least theoretically). Theorem 1 confirms this in a theoretical manner.
- In light of the above, I feel the contributions can be considered somewhat novel.
- However, there are several doubts/clarifications that pop in the mind of a reader (listed below) and clarification of them will help me judge the merits of this work in an informed manner.

__Quality__
- The overall technical quality of the paper seems reasonable. The supported claim is proven theoretically and also validated experimentally. Although experiments don't support it fully and I guess that is scope for future research.
- Authors are honest about articulating the weakness of the method observed during their experimentation.

__Clarity__

Overall, I felt that this paper lacks clarity and has ample scope for improving its writing. Below are some of the doubts/clarifications that struck me while reading the paper.

- In equation (2), I assume the matrix $M$ would be different for different coordinate axes as well as relations? If yes, you may want to use the symbol $M_r(k)$ or $M_{rk}$, where $k$ is the index of a coordinate axis and $r$ denotes the considered relation. Given that you have used index $k$ in the definition for the parameter $\theta_p(k)$, introducing $k$ for $M$ will make notations a bit more consistent. In fact, by the same token, matrices $S, S^{-1}$, as well as parameters $a$ and $b$ should also bear the index $k$ and $r$. Please correct me if I have misunderstood something here.
- By the same logic, even for Equation (3), you may want to consider using the symbol $p_{rk}$ or $p_r(k)$ if $M$ is indexed with $r$ and $k$ on the right side of the equation? If these notions become clumsy then you may want to add a disclaimer saying that you are omitting some indices to keep the notation less cluttered.
- While I could infer from reading that Rat-Pro model embeds entities into a complex space ${\cal{C}}^d$, would it make sense to state it explicitly somewhere?
- After reading the last line of the _Conclusion_ section, I am a bit confused as it seems to suggest that the current embedding is being performed in ${\cal{C}}^1$ space? Table 2 of the appendix seems to suggest that $d>1$. Please clarify this.
- In Equation (10), the outer summation is over variable $x$ which doesn’t show up in summand term. Can you make it clear?
- In equation (10), it's not clear what is $\mathbf{x}_j$ vector and what is its dimension? If running index $j$ corresponds to an axis then I believe $\mathbf{x}_j$ would be a scaler? But in that case, Hadamard product does not make sense.
- What benefits do you get by using the loss form given in Equation (10) as opposed to minimizing other simpler forms such as $\mathbf{x}_j \odot (1-\mathbf{x}_j)$ which also try to push the values of $\mathbf{x}_j$ towards $0$ or $1$?
- Given that the overall optimization problem becomes an integer programming problem ($a, b \in {0,1}$),  how do you finally ensure that in the learned matrix M, the diagonal terms $a$ and $b$ are exactly $0$ or $1$. I suppose the inclusion of loss term (10) still does not guarantee to have the learned values of the parameters $a,b$ being $0$ or $1$. Do you round off the learned values for $a$ and $b$?
- The projection matrix $M$ seems to be orthogonal projection matrix also because it appears that $S^{-1} = S^{\top}$ and hence $M=M^{\top}$. In this case, it may be better to explicitly state that you are focusing on the orthogonal projections of the entities.
-  In Figures 3(a), 3(b), and 3(c), what does the $y$-axis corresponds to?
- Sentence in line # 238 needs some rewriting as it doesn’t read properly.
- In Figure 4(a), how does the pattern for Hits@1 score look like? Does it follow the same trend as Figure 4(a)? For KG embeddings, I feel Hits@1 is crucial to track.
- How about the expressivity of the Rat-Pro model? Is there any relation that can’t be model by the proposed approach? Can you add some discussion around that?

__Significance__

- Overall, this paper could have a noticeable impact in advancing the SOTA provided it satisfactorily addresses the doubts/concerns raised above.
- The experimental section exposes a limitation saying that the presence of transitive relations dilutes the learning of other types of relations (as discussed in Section 4.5 and 4.6). These limitations kind of puts a question mark saying whether this paper managed to bridge the research gap completely. However, I feel, if not completely, this paper still managed to bridge the research gap and advance the SOTA to a decent and respectable level.

**Time Spent Reviewing:**

5

---

> ### Author Response · Authors · 2021-08-09
> **Response to Reviewer 99hH**
>
> Thank you for the careful reading and constructive feedback towards improving our manuscript. We address the reviewer's points below (numbered in order) and will update our manuscript accordingly:
>
> ***Q1:*** *In equation (2), I assume the matrix* $M$ *would be different for different coordinate axes as well as relations? If yes, you may want to use the symbol* $M_r(k)$ *or* $M_{rk}$, *where k is the index of a coordinate axis and r denotes the considered relation. Given that you have used index k in the definition for the parameter* $θ_p(k)$*, introducing* $k$ *for* $M$ *will make notations a bit more consistent. In fact, by the same token, matrices* $S$, $S^{-1}$, *as well as parameters a and b should also bear the index* $k$ *and* $r$​​​​. *Please correct me if I have misunderstood something here.*
>
> ***Q2:*** *By the same logic, even for Equation (3), you may want to consider using the symbol* $p_{rk}$​ *or* $p_r(k)$​ *if* $M$​ *is indexed with* $r$​ *and* $k$​ ​*on the right side of the equation? If these notions become clumsy then you may want to add a disclaimer saying that you are omitting some indices to keep the notation less cluttered.*
>
> **Re1&2:** Your understanding is correct. Thanks very much for your suggestion. We shall add a disclaimer to the paper to explain that we omit some indices in specific equation to make the notation less cluttered.
>
> ***Q3:*** *While I could infer from reading that Rot-Pro model embeds entities into a complex space* $C^d$, ​*would it make sense to state it explicitly somewhere?*
>
> **Re3:** We mentioned in the Preliminary that RotatE embeds entities to complex vector space. As an extension to RotatE, Rot-Pro naturally embeds entities to complex vector space and hence we do not state it explicitly. It would be more clear to state that Rot-Pro embeds entities into $C^d$, thanks for your suggestion.
>
> ***Q4:*** *After reading the last line of the Conclusion section, I am a bit confused as it seems to suggest that the current embedding is being performed in* $C^1$ *space? Table 2 of the appendix seems to suggest that* $d>1$*. Please clarify this.*
>
> **Re4:** In our model, the embedding space for entities is complex vector space $\mathbb{C}^d (d > 1)$, and *extending the complex space to higher dimension space* here means that the complex space $\mathbb{C}$ could be further extended to higher dimension space like the quaternion space $\mathbb{H}$ as the cited works [28, 21]. For instance, the embedding space for entities would become $\mathbb{H}^d$. We will rewrite the sentence to make it more clear.
>
> ***Q5:*** *In Equation (10), the outer summation is over variable x which doesn't show up in summand term. Can you make it clear?*
>
> ***Q6:*** *In equation (10), it's not clear what is* $x_j$​ *vector and what is its dimension? If running index* $j$ ​*corresponds to an axis then I believe* $x_j$ ​*would be a scaler? But in that case, Hadamard product does not make sense.*
>
> **Re5&6:** In the outer summation $\sum_{x \in \{a, b\}}$​, we use $x \in \{a, b\}$​ to represent that $\mathbf{x}$​ can be either the vector $\mathbf{a}$​ or $\mathbf{b} \in \mathbb{R}^d$​​. The outer summation actually means that
> $$
> L_{p} =\sum_{j=1}^{|R|}( ||(\mathbf{a}_j - 1.0) \odot \mathbf{a}_j \odot \mathbf{q}_j ||_2 + || (\mathbf{b}_j - 1.0) \odot \mathbf{b}_j \odot \mathbf{q}_j ||_2),
> $$
>
> where $|R|$ is the number of relations. We noted that there is a typo in Equation (10), where the inner summation should be a summation over $|R|$, rather than the dimension of the embedding $d$. Sorry for causing confusion, and we will fix it in the revised version, thanks.
>
> ***Q7:*** *What benefits do you get by using the loss form given in Equation (10) as opposed to minimizing other simpler forms such as* $x_j \odot (1−x_j)$​ *which also try to push the values of* $x_j$​ *towards* $0$​ *or* $1$​​ *?*
>
> **Re7:** When simply take $\mathbf{a}_j \odot (1−\mathbf{a}_j)$​ for optimizing, the final $\mathbf{a}_j$​ may not converge close enough to 0 or 1, and there may have a larger deviation (similar analysis applies to $\mathbf{b}_j$). In order to make the deviation smaller, we introduce a learnable weight $\mathbf{q}_j$​ as a penalty to make $\mathbf{a}_j$​ converge closer to $0$​ or $1$​, i.e. assign a value $\beta > 1$​ to $\mathbf{q}_j$​ when the difference between $\mathbf{a}_j$​ and $0$​ or $1$​ are larger than a vary small value $\gamma_m$​. We have also performed a set of comparative experiments to verify its effectiveness. We train Rot-Pro with simplified loss as you proposed $\mathbf{a}_j \odot (1−\mathbf{a}_j)$​, which is denoted as Rot-Pro-simple, and compare it with the Rot-Pro with our loss.
>
> For Rot-Pro,  the largest deviation of elements in $\mathbf{a}_j$ to 0 or 1 is just **5.9e-7**. The overall SSE (the sum of squares due to error) value is **6.9e-11**.
>
> For Rot-Pro-simple, the largest deviation is **0.08**. The overall SSE value is **0.012**.
>
> ***Q8:*** *Given that the overall optimization problem becomes an integer programming problem* $(a,b\in 0,1)$​​​,  *how do you finally ensure that in the learned matrix M, the diagonal terms a and b are exactly* $0$ ​​​*or* $1$. ​​​*I suppose the inclusion of loss term (10) still does not guarantee to have the learned values of the parameters a,b being* $0$​ *or* $1$​. *Do you round off the learned values for* $a$ *and* $b$​​​?
>
> **Re8:** We did not round off the learned values of $a$​ and $b$​. As we stated in Re7, the penalty we introduced in loss term (10) has already push the learned values of a and b closer enough toward $0$​ or $1$​ with the overall SSE value 6.9e-11. Hence we use the learned values directly for inference.
>
> ***Q9:*** *The projection matrix* $M$ *seems to be orthogonal projection matrix also because it appears that* $S^{-1}=S^{T}$ *and hence* $M=M^{T}$​​. *In this case, it may be better to explicitly state that you are focusing on the orthogonal projections of the entities.*
>
> **Re9:** Yes. Although we did not state explicitly, the idempotent projections used in this paper are essentially orthogonal projections. We will mention it in the revised version.
>
> ***Q10:*** *In Figures 3(a), 3(b), and 3(c), what does the y-axis corresponds to?*
>
> **Re10:** The y-axis corresponds to the number of elements (dimensions) with a specific relational rotation phase $0 \leq\theta\leq 2\pi$ in embedding vectors of all relations (the total number of elements is $|R|\times d$) with the precondition that the corresponding projection is non-trivial (i.e. parameters a and b should be 1, 0 or 0, 1). For example, among dimensions that have non-trivial projection, there are 3600 elements (dimensions) has a relational rotation phase $\pi$.
>
> ***Q11:*** *Sentence in line # 238 needs some rewriting as it doesn't read properly.*
>
> **Re11:** Thanks for pointing out. We rewrite it as "we construct three sub-test sets S1, S2, S3 of YAGO3-10 for evaluation, which consist of a single transitive relation *isLocatedIn.*"
>
> ***Q12:*** *In Figure 4(a), how does the pattern for Hits@1 score look like? Does it follow the same trend as Figure 4(a)? For KG embeddings, I feel Hits@1 is crucial to track.*
>
> **Re12:** The result of Hit@1 on S1, S2, S3 test sets are listed as follows:
>
> |                                                         | S1    | S2    | S3    |
> | ------------------------------------------------------- | ----- | ----- | ----- |
> | **RotatE**                                                  | 21.65 | 26.01 | 35.06 |
> | **Rot-Pro-**$(-\pi, \pi)$**-Init**                      | 23.48 | 26.22 | 34.07 |
> | **Rot-Pro-**$(-\frac{\pi}{2}, \frac{\pi}{2})$**-Init**  | 21.16 | 30.63 | 38.89 |
> | **Rot-Pro-**$(-\frac{\pi}{2}, \frac{\pi}{2})$**-Train** | 24.4  | 30.9  | 54.43 |
>
> The pattern for Hits@1 score also follows the same trend as Hit@10.
>
> ***Q13:*** *How about the expressivity of the Rot-Pro model? Is there any relation that can't be model by the proposed approach? Can you add some discussion around that?*
>
> **Re13:** As proved in the paper, Rot-Pro can express five common relation patterns:  symmetry, asymmetry, inversion, composition and transitivity. There are still other  relation patterns defined in OWL2 Web Ontology Language. Some of them can be easily expressed by common knowledge graph embedding methods and others are actually constraints imposed on models which are irrelevant to the expressiveness. For instance, a relation pattern *functionality* defined in OWL2 requires its instances follow the rule that: $r$ is a functional relation, if $(a, r, b)$ is an instance of $r$ implies that for any $c \ne b$, $(a, r, c)$ is not an instance of $r$. Most geometric knowledge graph embedding models can express such relation. However, more investigation is required to prevent mis-prediction like $(a, r, c)$, which is beyond the scope of our paper.
>
> For further discussion, if we extend the discussion beyond common relation patterns. We believe that Rot-Pro has the same problem as many other models that they cannot express 1-To-N relations exactly. Although these models yield relatively good performance, they did not theoretically solve the 1-To-N problem and just map them adjacently in vector space. Therefore, enabling knowledge graph models to exactly express 1-To-N relations may become a future research direction.

---

> > ### Comment · Reviewer_99hH · 2021-08-28
> > **Score updated post rebuttal (5->6)**
> >
> > Dear Authors,
> > Thanks for your response. My doubts are not clear and I am changing my rating.

---

### Official Review · Reviewer_NmZc · 2021-07-17

**Rating:** 7
**Confidence:** 3

**Summary:**

Summary: This study proposes a new geometric model Rot-Pro that enables the learning of transitive inference patterns by combining rotation and projection. The core idea is to involve a learnable idempotent projection onto a reduced basis. This achieves the transitivity pattern without forcing the entity embeddings to be the same as it is in previous models. The study shows the effectiveness in experimental results mostly on the small Countries dataset, but also shows that the model performs well on larger KG benchmarks especially on the Yago3-10 dataset. The authors also report their investigation into initialization schemes that are required for successfully training the model.

**Limitations And Societal Impact:**

The limitations and societal impact haven been  addressed.

**Main Review:**

Originality: There is a lot of recent work in geometric models that explicitly investigate modelling certain inference patterns and this study proposes to investigate transitivity in a rotational model that expands its capabilities to more inference patterns. I was missing a comparison and discussion to BoxE, which is a strong model in this area.

Quality: The line of arguments and to motivate the study are well laid out, and proofs and experiments support the claim. The authors point out the challenge of initialization for the model and they mention a special loss to enforce zeros and ones in the projection. Did the authors also consider something like [1]? In the experimental results I was also missing a comparison to BoxE, which reports better results on Yago3-10.

Clarity: The paper was written very well and has a nice presentation.

Significance: The paper presents a solid idea which is executed well to show the effectiveness of explicitly allowing a model to learn transitive patterns in the data. While the authors note that there is room for improvement, this does not limit the usefulness of the analysis, proposed model and results.

[1] Louizos, C., Welling, M., and Kingma, D. P. Learning sparse neural networks through l0 regularization. CoRR, 2017.


**Time Spent Reviewing:**

1.5

---

> ### Author Response · Authors · 2021-08-09
> **Response to Reviewer NmZc**
>
> Thank you for recognizing our work as a solid idea and providing helpful suggestions. Below you will find our responses to your comments.
>
> ***Q1:*** *I was missing a comparison and discussion to BoxE, which is a strong model in this area. And in the experimental results missing a comparison to BoxE, which reports better results on Yago3-10.*
>
> **Re1**: BoxE is indeed a representative work recently. It embeds entities as points, and relations as a set of boxes, for yielding a *fully expressive* model. BoxE is an extension on trans-series models, while our method could be regarded as an extension on linear mapping models. The *fully expressive* of BoxE refers to that it is able to express inference patterns, which includes symmetry, anti-symmetry, inversion, composition, hierarchy, intersection, and mutual exclusion. However, it does not model and infer the transitivity pattern. We also conducted experiments on the three sub-test sets S1, S2, S3 we sampled from YAGO3-10 as described in the paper to verify this. The experimental results are listed as below:
>
> Result on S1 test set with [MRR, Hit@1, Hit@3, Hit@10]:
>
> |             | MRR   | Hit@1 | Hit@3 | Hit@10 |
> | ----------- | ----- | ----- | ----- | ------ |
> | **BoxE**    | 0.343 | 0.255 | 0.385 | 0.504  |
> | **Rot-Pro** | 0.337 | 0.247 | 0.376 | 0.512  |
>
>
> Result on S2 test set on [MRR, Hit@1, Hit@3, Hit@10]:
>
> |             | MRR   | Hit@1 | Hit@3 | Hit@10 |
> | ----------- | ----- | ----- | ----- | ------ |
> | **BoxE**    | 0.290 | 0.262 | 0.291 | 0.342  |
> | **Rot-Pro** | 0.328 | 0.235 | 0.366 | 0.522  |
>
> Result on S3 test set on [MRR, Hit@1, Hit@3, Hit@10]:
>
> |             | MRR   | Hit@1 | Hit@3 | Hit@10 |
> | ----------- | ----- | ----- | ----- | ------ |
> | **BoxE**    | 0.381 | 0.349 | 0.385 | 0.439  |
> | **Rot-Pro** | 0.447 | 0.337 | 0.517 | 0.626  |
>
> The results show that BoxE is an expressive model with comparable performance on common evaluation metrics as Rot-Pro. However, the results of Rot-Pro on S2 and S3 outperforms BoxE, which illustrates that the advantage of Rot-Pro of modeling transitivity becomes obvious when the hops of transitivity grows.
>
> We would surely add a comparison to BoxE in related work as well as experimental results in a revised version, thanks.
>
> ***Q2:*** *The authors point out the challenge of initialization for the model and they mention a special loss to enforce zeros and ones in the projection. Did the authors also consider something like [1]?*
>
> **Re2**: Thanks for pointing out Reference [1]. [1] uses a re-parameterization to learn a variable with 0/1 constraints, which could be an interesting solution for our problem. We will consider it in the future, thanks.

---

### Official Review · Reviewer_uEYp · 2021-07-17

**Rating:** 6
**Confidence:** 4

**Summary:**

This paper noticed that if a relation projected orthogonally to the translation direction, it becomes transitive, and then make this insight work with other relational patterns.


**Ethical Concerns:**

No ethical issues.

**Limitations And Societal Impact:**

Yes. Section 4.6 was nice to see!

**Main Review:**

I liked this paper. It had a good insight and seems to work well. It has good originality, and is mostly clear. It is potentially significant.

However, I am very suspicious of whether there are side effects. For example, ancestor is the transitive closure of parent. Because ancestor has to have some particular form in order to be transitive, doesn't that imply that parent must also have some particular form?  What happens when parent is used in other relations? While I believe your theorem (although I didn't check the related papers), it doesn't show that they can be handled simultaneously. If there was also a modularity result, the consequent of the design patterns do not affect the representation of the condition  of the design pattern  (i.e., they work for all representations of the condition), I would be more convinced of your claim that you can handle all relation patterns. (Claiming to handle all is ambiguous as to whether you can handle them separately or together.)

Isn't transitivity an instance of composition, with r = r \circle r?  I think there are more assumptions behind the design patterns than you state.

Do your results imply other patterns, eg  r = r1 \circle r?  Are there other patterns that can exist? You should tell us whether this is the end of the story of representing design patterns of whether there are more. (Students looking for research problems want to know).

I very much appreciated section 4.6 on limitations.

Please label x and y axes of all figures. Figure 5 b has loss on the y-axis but how can it be negative?



**Time Spent Reviewing:**

3

---

> ### Author Response · Authors · 2021-08-09
> **Response to Reviewer uEYp**
>
> Thank you for liking our paper and thoughtful review. Below is our responses to your comments.
>
> ***Q1:*** *For example, ancestor is the transitive closure of parent. Because ancestor has to have some particular form in order to be transitive, doesn't that imply that parent must also have some particular form? What happens when parent is used in other relations?*
>
> **Re1:** As you said, ancestor is the transitive closure of parent. According to the definition, the transitive closure of transitive relation is the relation itself. So ancestor is a transitive relation, while parent is not a transitive relation. They are different relations with different properties. So they are embeded independently in current knowledge graph embedding models. The transitivity of ancestor does not imply parent should also have the same particular form.
>
> ***Q2:*** *Can the model handle all relation patterns simultaneously?*
>
> **Re2:** This is an interesting question and worth deeper discussion. In the many previous works, the proof of expressiveness is focused on the expressiveness of each relation pattern. Following the same way, our proof in the paper only proves that each relation pattern can be expressed with Rot-Pro model. It still needs more investigation to see that whether all relation patterns can be expressed simultaneously considering the complex interactions between different relation patterns in knowledge graphs.  Furthermore,  what if a single relation exhibits multiple relation patterns? In that case, we think most current models cannot handle this effectively. Our preliminary analysis found that the Rot-Pro model can express relations that are both transitive and symmetric, e.g. the relation *isEqualTo.* We believe that Rot-Pro has the potential to become a model that can express all relation patterns simultaneously with further extensions.
>
> ***Q3:*** *Isn't transitivity an instance of composition with r = r \circle r? I think there are more assumptions behind the design patterns than you state.*
>
> **Re3:** $r = r \circ r$​ ​is an equivalent definition of that $r$​​ has transitivity. However, transitivity is not an instance of composition. In knowledge graph, composition is usually used to define a new relation $r$​​ as the composition of some other relations $r_1, r_2, \ldots, r_n$​​, i.e. $r = r_1 \circ r_2 \circ \cdots \circ r_n$​​, where $r$​​ is different from $r_i$ ​​($1 \leq i \leq n$​​).
>
> ***Q4:*** *Do your results imply other patterns, eg r = r1 \circle r? Are there other patterns that can exist? You should tell us whether this is the end of the story of representing design patterns of whether there are more. (Students looking for research problems want to know).*
>
> **Re4:** In fact, all five typical relation patterns have been discussed and can be theoretically modeled separately by Rot-Pro. However, it is definitely not the end of the story. As the cases we discussed in Re2, it is very interesting to design a model which can handle all common relation patterns simultaneously. There are also other relation patterns used in semantic web, which are not currently supported by knowledge graph embedding models. We will add these discussions in the revised version.
>
> ***Q5:*** *Please label x and y axes of all figures.  Figure 5 b has loss on the y-axis but how can it be negative?*
>
> **Re5:**  We will make a clear description of labels of x and y axes in the caption of figures in the revised version. Thanks for pointing out.
>
> Strictly speaking, the y-axis in Figure 5 b is actually not the loss defined in the Eq (9) but a simplified version which only takes the $-(\gamma-d_{r}(\mathbf{h}, \mathbf{t}))$​ part in Eq (9), which is in direct proportion to the positive part loss $-\log \sigma(\gamma-d_{r}(\mathbf{h}, \mathbf{t}))$​ in Eq (9). When the relational rotation phase is $2n\pi$​, the distance between head and tail entities is small, thus causing the result to be negative. We will add detailed explanation to the figure to make it more clear for the readers. Thanks for pointing out.

---

> > ### Comment · Reviewer_uEYp · 2021-08-31
> > **further comment**
> >
> > I am not sure "So they are embeded independently in current knowledge graph embedding models" is true. They can't embed independently; ancestors can only progress by parent relations. It is not clear to me that the interaction works (as you discuss later). It is not much use being able to represent transitivite relations if you can't seed them by non-trasititive relations, as in the ancestor parent example). Please include a discussion of this in the paper.
> >
> > Also the limitations should be much clearer and up-front. Tell us what you can't do (being clear about what you know you can't do and what you suspect you can't do). I think it should even be in the abstract, but that is up to you.

---

> > > ### Author Response · Authors · 2021-09-01
> > > **Re: further comment**
> > >
> > > The embedding methods considered in this paper, such as TransE and RotatE, treat different relations in the knowledge graph independently. That is, the interactions between relations in these methods have the possibility to be learned from the data, but they are not explicitly modeled (current benchmarks for evaluation even do not provide the ontology that describe the relationship between relations).
> > >
> > > Does the comment "*ancestors can only progress by parent relations. It is not clear to me that the interaction works (as you discuss later). It is not much use being able to represent transitive relations if you can't seed them by non-transitive relations, as in the ancestor parent example).*" mean that: the learned embeddings of ancestor and parent relations satisfy the constraint that any two entities have the parent relation implies they also have the ancestor relation?
> > >
> > > If yes, current embedding methods are only simplified versions of the way described in your comment that the complex semantics between relations is explicitly considered. It should be a further research direction of knowledge graph embedding. We will update our paper with a further discussion.

---

### Official Review · Reviewer_VTBj · 2021-07-19

**Rating:** 7
**Confidence:** 4

**Summary:**

When embedding entities and relations in knowledge graphs, operations such as translations, matrix-vector products, quadratic forms, etc are used to represent relations between entity vectors. While these vector-based methods are flexible, they do not specifically encode an inductive bias towards transitive relations that is necessary for modeling certain hierarchies. This paper proposes an approach based on projections (idempotent transformations) and rotations taken together, which allows it to exactly model transitivity in relations.

**Limitations And Societal Impact:**

Limitations with respect to local optima in optimization, and initialization, are discussed extensively in the experiments.

**Main Review:**

Update: I will stick with my current score, I believe that the authors have adequately enough addressed reviewers' comments.

**Originality**

There has been much work on modeling transitive relations with specialized embedding techniques such as order embeddings, hyperbolic embeddings, box embeddings etc. However, integrating these approaches into standard multirelational / hypergraph KG embedding based on vectors, which allows for multiple heterogeneous relations to be embedded, is an open problem. As the authors note, the recent RotatE model is able to model (a)symmetry, composition, and inversion, but cannot specifically guarantee transitivity. They add a learned subspace projection step to the RotatE model and prove this can represent transitive relations.

**Quality**

The theoretical argument is sound. The evaluation is reasonable and standard for KG embedding techniques, and showing the model can indeed infer transitivity is welcome. The discussion of limitations and local optima in the model is welcome.

**Clarity**

The paper is mostly well written and argued. The final error-analysis section could use some light editing to tighten it up, as it currently reads more like an appendix discussion.

**Significance**

The RotatE model is simple and powerful, and this extension of it seems similarly simple and powerful. Knowledge base embedding approaches are popular and this might become a new baseline technique.


**Time Spent Reviewing:**

1

---

> ### Author Response · Authors · 2021-08-09
> **Response to Reviewer VTBj**
>
> Thank you for your recognition for our paper: "The RotatE model is simple and powerful, and this extension of it seems similarly simple and powerful. Knowledge base embedding approaches are popular and this might become a new baseline technique." Below is our responses to your comments.
>
> ***Q1:*** *The final error-analysis section could use some light editing to tighten it up, as it currently reads more like an appendix discussion.*
>
> **Re1:** Thanks for the advice, and we will tighten this section up in the revised version. Also thanks for the recognition for the Rot-Pro, and we also expect it to become a simple and powerful baseline model in the future.

---

### Decision · Program_Chairs · 2021-09-27

**Decision:**

Accept (Poster)

**Comment:**

the reviewers agreed that the paper makes a solid contribution by proposing a model that can infer transitivity, together with convincing experimental results. Most of the reviewers' concerns have been cleared in the discussion phase. The authors should make sure to incorporate the necessary updates in the camera-ready version.